# Role of the WNT/β-catenin/ZKSCAN3 Pathway in Regulating Chromosomal Instability in Colon Cancer Cell lines and Tissues

**DOI:** 10.3390/ijms23169302

**Published:** 2022-08-18

**Authors:** Young-Eun Cho, Jeong-Hee Kim, Young-Hyun Che, Yong-Jun Kim, Ji-Youn Sung, Yoon-Wha Kim, Bong-Geun Choe, Sun Lee, Jae-Hoon Park

**Affiliations:** 1Department of Pathology, College of Medicine, Kyung Hee University, Seoul 02453, Korea; 2Department of Pathology, Kyung Hee University Hospital, College of Medicine, Kyung Hee University, Seoul 02447, Korea; 3Department of Preventive Medicine, College of Medicine, Kyung Hee University, Seoul 02453, Korea

**Keywords:** ZKSCAN3, MAD2L2, WNT, β-catenin, colon cancer

## Abstract

Zinc finger protein with KRAB and SCAN domains 3 (ZKSCAN3) acts as an oncogenic transcription factor in human malignant tumors, including colon and prostate cancer. However, most of the ZKSCAN3-induced carcinogenic mechanisms remain unknown. In this study, we identified ZKSCAN3 as a downstream effector of the oncogenic Wnt/β-catenin signaling pathway, using RNA sequencing and ChIP analyses. Activation of the Wnt pathway by recombinant Wnt gene family proteins or the GSK inhibitor, CHIR 99021 upregulated ZKSCAN3 expression in a β-catenin-dependent manner. Furthermore, ZKSCAN3 upregulation suppressed the expression of the mitotic spindle checkpoint protein, Mitotic Arrest Deficient 2 Like 2 (MAD2L2) by inhibiting its promoter activity and eventually inducing chromosomal instability in colon cancer cells. Conversely, deletion or knockdown of ZKSCAN3 increased MAD2L2 expression and delayed cell cycle progression. In addition, ZKSCAN3 upregulation by oncogenic WNT/β-catenin signaling is an early event of the adenoma–carcinoma sequence in colon cancer development. Specifically, immunohistochemical studies (IHC) were performed using normal (NM), hyperplastic polyps (HPP), adenomas (AD), and adenocarcinomas (AC). Their IHC scores were considerably different (61.4 in NM; 88.4 in HPP; 189.6 in AD; 246.9 in AC). In conclusion, ZKSCAN3 could be responsible for WNT/β-catenin-induced chromosomal instability in colon cancer cells through the suppression of MAD2L2 expression.

## 1. Introduction

Catenin-β is a multifunctional intracellular signaling protein that plays a critical role in cancer development and the maintenance of cellular homeostasis by regulating gene transcription [1]. In the absence of Wnt factors, β-catenin is effectively degraded by a multiprotein complex composed of APC, Axin1, GSK3β, and CK1 [2,3]. Upon stimulation by the Wnt ligand, β-catenin is rescued; it accumulates in the cytoplasm, and is transported to the nucleus, where it interacts with T-cell factor/lymphocyte enhancer factor (TCF/LEF) to activate the transcription of target genes involved in cell cycle progression and cell survival [4]. The upregulation of β-catenin, frequently caused by *APC* mutations, occurs in the early stages and is one of the most common events in the development of colorectal cancer [5]. In addition, chromosomal instability (CIN), observed in 65−70% of sporadic colorectal cancers, can be induced by the activation of β-catenin signaling [6].

CIN, caused by impaired DNA repair, improper chromosomal segregation, or spindle checkpoint defects during cell division, is commonly observed in human colon cancer [7,8]. CIN is not only a crucial event for tumor initiation or progression toward malignancy, but also a principal driver of genetic heterogeneity and clonal diversification; therefore, it is closely associated with the aggressive behavior of cancer cells, resistance to chemotherapy, and poor survival in colon cancer patients [9,10,11]. This makes elucidating the molecular mechanisms underlying CIN in colon cancers very important for understanding the carcinogenic process and developing better therapeutic strategies. Aberrant activation of the Wnt signaling pathway is one of the most important contributors to the development of colon cancers; it induces CIN and allows the β-catenin-mediated rapid and continuous proliferation of cells harboring CIN [12,13]. β-catenin/TCF-mediated transcription increases CIN through dysregulation of G2/M progression or chromosome segregation errors [14]. Activated WNT triggers defects in kinetochore–microtubule attachment during mitosis, leading to inaccurate duplication of genetic material [15]. *APC* mutation and overexpression of Conductin or Aurora A are downstream effectors responsible for CIN induction by WNT pathway activation [12,16,17]. However, further studies on the role of chromosomal segregation errors in CIN induction, knowledge of the molecules linking the spindle assembly checkpoint (SAC) and activated WNT/β-catenin pathway are necessary.

Zinc finger with KRAB and SCAN domain 3 (ZKSCAN3) is a member of the zinc finger transcription factors with Krüppel-associated box (KRAB) and SCAN domains. It participates in a variety of critical cellular functions, including autophagy, cell differentiation, apoptosis, cell proliferation, and cancer progression [18,19,20,21]. ZKSCAN3 was initially identified as a driver of colorectal cancer; however, it also plays an important role in cancer cell proliferation, migration, and metastasis in several human malignancies, including colon, bone marrow, prostate, bladder, and cervical cancers [21,22,23,24,25,26]. Furthermore, the upregulation of ZKSCAN3 increases the expression of colon cancer progression factors such as cyclin D1, epidermal growth factor receptor, integrin β4, and vascular endothelial growth factor [27]. However, the molecular mechanisms underlying cancer development/progression in response to ZKSCAN3 overexpression and its downstream signaling molecules are yet to be elucidated.

In this study, we investigated the molecular mechanisms and pathways underlying colon cancer development mediated by ZKSCAN3 in relation to the oncogenic WNT pathway, because ZKSCAN3 and WNT/β-catenin pathways are well known in colon carcinogenesis. We demonstrated that the WNT/β-catenin/ZKSCAN3 axis could act as a transcriptional repressor of *MAD2L2* and that dysregulation of ZKSCAN3 could induce CIN through a MAD2L2-dependent pathway. This is the first study to show that ZKSCAN3 is a key mediator of CIN induction in the oncogenic WNT/β-catenin pathway.

## 2. Results

### 2.1. Identification of the ZKSCAN3 Target Gene Pathway through Transcriptome Analysis

ZKSCAN3 plays a significant role in cervical cancer progression [26]. To gain further insight into the molecular signaling pathways regulated by ZKSCAN3, homozygous (*ZK3*-KO) and heterozygous (*ZK3*-KD) *ZKSCAN3*-deficient HCT116 cell lines were constructed using the CRISPR/Cas9 system (Appendix A). RNA sequencing was used to compare gene expression in the parental HCT116 cells and *ZK3*-KO cells. Heatmap analysis (Figure 1A) indicated that 1923 genes with significant changes in expression levels (FDR < 0.05) were clustered; among them, 1069 genes were upregulated in *ZK3*-KO cells and 854 were downregulated. In addition, gene ontology (GO) and enrichment analysis using the DAVID software identified 64 canonical pathways that were significantly altered due to ZKSCAN3 deficiency (Figure 1B). In *ZK3*-KO cells, the significantly upregulated differentially expressed genes (DEGs) (*p* < 0.05) included genes enriched in biological processes such as positive regulation of the mitotic cell cycle spindle assembly checkpoint and those involved in the G1/S transition of the mitotic cell cycle (Figure 1B). In addition, the expression of genes that play an essential role in the WNT signaling pathways was markedly altered in the *ZK3*-KO cells. Therefore, the association between ZKSCAN3 and WNT signaling pathways was investigated. Deep sequencing results showed that ZKSCAN3 was involved in cell cycle regulation; therefore, we used the KEGG and STRING algorithms to identify the specific biological processes closely related to genes classified into the cell cycle categories. The genes recognized as cell cycle-related factors (Figure 1C and Appendix A) participate in functional networks during cell division, such as anaphase-promoting complex-dependent catabolic processes, Mad3/BUB1 homology region 1, centralspindlin complexes, chromosome passenger complex, cohesin loading onto chromatin, mitotic cytokinesis, and G2/M DNA replication checkpoint (Figure 1D). To confirm the results of gene expression profiling, we validated the transcriptional levels of selected genes involved in mitotic cytokinesis mechanisms and checkpoint regulation (Figure 1E). The qRT-PCR analysis confirmed that *ZKSCAN3* deficiency upregulated these genes (Figure 2). The RNA sequencing results suggest that ZKSCAN3 could play a role in tumorigenesis by inhibiting the spindle assembly checkpoint pathway and preventing improper chromosomal segregation.

### 2.2. Activation of the WNT Pathway Induces ZKSCAN3 Upregulation

Ontology analysis indicated a possible role of ZKSCAN3 in the regulation of the cell cycle and Wnt signaling pathway (Figure 1B). We focused on the role of β-catenin in the Wnt signaling pathway. We compared the deep sequencing data of *ZK3*-KO with that of *CTNNB1*-KD HCT116 cells using a published dataset (GSE95670). The genes with altered expression in both *ZK3*-KO and *CTNNB1*-KD HCT116 cells were involved in the regulation of the WNT signaling pathway, mitotic sister chromatid segregation, microtubule-based processes, and cell cycle control (Figure 3A,B). We evaluated whether the oncogenic ZKSCAN3 and WNT pathways function individually or in concert to develop CRC. HCT116 cells were treated with human recombinant WNT factors (WNT-1, WNT-3A, and WNT-5A) or the GSK-3 inhibitor (CHIR 99021); ZKSCAN3 was upregulated by WNT pathway activation (Figure 3C). In addition, ZKSCAN3 upregulation by Wnt factor proteins was observed in the normal colonic epithelial-derived cells, NCM365D (Appendix A). PNU-74654 (PNU) is a WNT pathway inhibitor, which prevents the interaction between β-catenin and TCF4 [28]. To clarify the mechanism of ZKSCAN3 upregulation by WNT pathway activation, we performed qPCR and Western blotting for ZKSCAN3 following WNT-3A treatment in the presence or absence of PNU using HCT116 cells. PNU inhibited the effect of WNT-3A on ZKSCAN3 upregulation (Figure 3D,E). Therefore, WNT-3A upregulated ZKSCAN3 via the WNT/β-catenin pathway. To clarify the role of β-catenin in the regulation of ZKSCAN3 expression, we performed reporter assays. HCT116 cells were transfected with both pGL-ZK3 and a β-catenin expression plasmid (pCTNNB1) or empty plasmid for 24 h. The cells were treated with WNT-3A with or without PNU or MSAB to promote β-catenin degradation [29]. Luciferase assays indicated that ZKSCAN3 upregulation by Wnt pathway activation was dependent on the integrity of β-catenin/TCF4 (Figure 3F). Promoter analyses indicated β-catenin/TCF4 regulation of the ZKSCAN3 promoter; therefore, we evaluated whether β-catenin or TCF-4 could bind directly to the ZKSCAN3 promoter. Candidate TCF4 binding motifs were identified in the human genome sequence of the ZKSCAN3 promoter (Appendix A) [30]. We designed PCR primers targeting this site in the ZKSCAN3 promoter and performed chromatin immunoprecipitation (ChIP) assays for the TCF4 protein. TCF4 binding to the ZKSCAN3 promoter was observed in the GFP-TCF4 plasmid (pGFP-TCF4)-transfected cells (Figure 3G,H). However, β-catenin did not bind to the ZKSCAN3 promoter (data not shown). Therefore, ZKSCAN3 could be a transcriptional target of the β-catenin/TCF4-dependent Wnt pathway.

### 2.3. ZKSCAN3 Deficiency Delays Mitosis Progression

ZKSCAN3 could be involved in cell cycle regulation or mitosis progression associated with the WNT/β-catenin pathway (Figure 1 and Figure 2). To clarify the relationship between ZKSCAN3 expression and cell cycle progression, we investigated whether ZKSCAN3 deficiency or downregulation affected the cell cycle profile under WNT factor stimulation. The cells were cultured with or without WNT-3A, and the cell proliferation rate was measured. The proliferation rate of *ZK3*-KO and *ZK3*-KD cells was substantially lower than that of *ZK3*-WT cells. WNT-3A treatment failed to rescue cell proliferation and invasiveness in *ZK3*-KO and *ZK3*-KD cells. This could explain why cell proliferation or invasive growth was dependent on the expression level of ZKSCAN3, and not that of WNT-3A (Figure 4A and Appendix A). The level of ZKSCAN3 expression was important for rapid cell proliferation and invasive growth of CRC in response to WNT pathway activation. Next, we analyzed the cell cycle profiles using flow cytometry to further understand the reduction in the proliferation rate in the *ZK3*-KO cells. Consistent with the decrease in cell proliferation rates, there was a marked increase in the G2/M phase of the cell cycle in *ZK3*-KO and *ZK3*-KD cells (Figure 4B). Nuclear envelope breakdown (NEBD) occurs in the late prophase and involves disassembly of the nuclear pore complex, depolymerization of the nuclear membrane, and removal of the nuclear membrane from chromatin [31]. Chromosomal segregation occurs during anaphase and telophase. Therefore, we analyzed the prometaphase–telophase (PT) time, which is from nuclear envelope breakdown (NEB) in prophase to centrosome segregation in telophase. *ZK3*-KO, -KD, and -WT cells were stably transfected with an RFP-tagged histone 2B-expressing plasmid (pRFP-H2B), and PT time was measured using a live cell imaging system. Approximately 78% of *ZK3*-WT cells had a PT time of less than 60 min (mean time: 36 min), whereas 50% and more than 60% of *ZK3*-KD and *ZK3*-KO cells, respectively, had PT times longer than 60 min (mean time: 64 and 72 min, respectively) (Figure 4C). These findings indicated a longer duration of mitosis progression in *ZKSCAN3*-knockdown or -deleted cells. Further analysis of the mitotic index by counting the metaphase cells demonstrated that the number of cells in the mitotic phase of the cell cycle increased remarkably in *ZK3*-KO cells compared with that in *ZK3*-WT cells (Figure 4D). Therefore, *ZKSCAN3* deficiency suppressed cell cycle progression in HCT116 colon cancer cells by delaying mitosis progression from prometaphase to telophase.

### 2.4. ZKSCAN3 Acts as a Transcriptional Repressor of Spindle Checkpoint Protein MAD2L2

To further investigate the molecular mechanisms underlying delayed mitosis progression in *ZK3*-KO cells, we used the EpiTect^®^ ChIP qPCR array Human Cell Cycle to identify transcription factors involved in cell cycle regulation. HCT116 cells were transduced using adenovirus expressing ZKSCAN3 tagged with HA (Ad-ZK3/HA). The DNA-protein complexes precipitated using an anti-HA antibody were subjected to qPCR assay. We identified gene promoters that interacted with ZKSCAN3 (Figure 5A). Among them, MAD2L2 is a member of the mitotic spindle assembly checkpoint and prevents improper chromosome segregation. It was strongly upregulated in *ZK3*-KO cells. ChIP analysis of individual gene promoters with the ZKSCAN3 protein confirmed the interaction between the ZKSCAN3 and *MAD2L2* promoters (Figure 5B). In addition, to investigate the effect of ZKSCAN3 on MAD2L2 expression, HCT116 cells were transfected with a GFP-tagged ZKSCAN3 expression plasmid (pZK3-GFP), and the mRNA and protein levels of MAD2L2 were examined. We noted a significant downregulation of both the mRNA and protein levels of MAD2L2 in ZKSCAN3-overexpressing cells. In addition, we observed the upregulation of MAD2L2 in *ZK3*-KD and *ZK3*-KO cells compared to that in *ZK3*-WT control cells (Figure 5C,D). These findings were further supported by a luciferase assay; the promoter activity of *MAD2L2* was enhanced in *ZK3*-KO cells, but suppressed in ZKSCAN3-overexpressing cells (Figure 5E). Therefore, ZKSCAN3 could act as a transcriptional repressor of MAD2L2.

### 2.5. Upregulation of ZKSCAN3 Induces Chromosomal Instability by Suppressing MAD2L2

Aberrant activation of the Wnt/β-catenin signaling pathway induces CIN [12,14], a critical step in colon carcinogenesis. ZKSCAN3 is a transcriptional repressor of certain spindle checkpoint proteins, which ensure proper chromosomal segregation. Therefore, we hypothesized that ZKSCAN3 upregulation could lead to CIN through WNT/β-catenin activation. ZKSCAN3-overexpressing HCT116 cell lines (#6/ZK3 and #16/ZK3) were established by stably transfecting pcDNAV5-ZK3 (Appendix A). After clonal establishment, cytokinesis-block micronucleus (CBMN) analysis was performed to detect nuclear abnormalities at passage numbers 40−50. The frequency of nuclear abnormalities, including micronucleus (MN), nuclear buds (NBUDs), and nucleocytoplasmic bridges (NCB), in ZKSCAN3-overexpressing cells was remarkably increased compared to that in the mock-transfected control cells (#4 N/mock) (Figure 6A). We further quantified chromosome copy numbers using FISH probes for chromosome-specific centromeric enumeration in interphase nuclei. The proportion of #4N cells with signals deviating from the modal values ranged from 2.7% (chromosome 7) to 3.6% (chromosome 17); however, the proportion of HCT116 cells with ZKSCAN3 overexpression ranged from 5.9% to 6.7% (#6/ZK3), and from 5.6% to 6.3% (#16/ZK3) (Figure 6B,C).

To investigate whether the CIN observed in ZKSCAN3-overexpressing cells was due to downregulation of MAD2L2, double ZKSCAN3- and MAD2L2-expressing cells (#17/ZK3/MAD and #19/ZK3/MAD) were constructed through stable co-transfection of pcDNAV5-ZK3 and pDEST-MAD2L2-GFP into HCT116 cells, and selection with double antibiotics (Appendix A). The cells with ectopic MAD2L2 expression exhibited a lower percentage of nuclear aberrations in CBMN analysis and chromosomal gain and loss in FISH compared to that in #6/ZK3 and #16/ZK3 cells (Figure 6A–C). In addition, downregulation of MAD2L2 by stable transfection with shRNA targeting MAD2L2 restored delayed mitotic progression and retarded cell proliferation (Appendix A). Therefore, aberrantly high ZKSCAN3 expression induced CIN by inhibiting MAD2L2 expression.

### 2.6. The WNT/β-Catenin/ZKSCAN3 Axis Acts during the Early Stages of Colon Cancer Development

Aberrant β-catenin expression caused by Wnt pathway activation and/or *APC* mutations plays a critical role in early carcinogenic processes [32,33,34]. ZKSCAN3 could participate in the carcinogenic process in the early stages of the adenoma–carcinoma sequence and could induce CIN; thus, functioning as a tumor progression factor. Therefore, we determined whether β-catenin-dependent ZKSCAN3 upregulation is observed in the early precursor lesions of colon cancer as well as in advanced cancers. We compared ZKSCAN3 expression by performing ZKSCAN3 immunohistochemical staining for normal colon epithelium, non-neoplastic hyperplastic polyps, adenomas, and advanced cancers. The IHC score of adenomas was substantially higher than those of normal colon mucosa or non-neoplastic hyperplastic polyps, but lower than that of advanced cancer (Figure 7A). In addition, ZKSCAN3 expression was more prominent in the dysplastic epithelia of the adenomas (Figure 7B). Therefore, ZKSCAN3 upregulation occurred in the early stages of colon cancer development. Increased gene copy numbers of ZKSCAN3 play a crucial role in ZKSCAN3 upregulation in cancers [21,23,26]. Therefore, we performed gene copy number variation analysis using qPCR to investigate whether gene copy variations occurred in the adenomas. Significant changes in the ZKSCAN3 gene copy numbers were observed only in advanced cancers and not in adenomas (Figure 7C). Therefore, the aberrant high expression of ZKSCAN3 in adenomas is not caused by genetic changes, unlike that in advanced cancers. We performed IHC using anti-ZKSCAN3 and anti-β-catenin antibodies in the contiguous sections of formalin-fixed paraffin-embedded (FFPE) tissues and examined the correlation between ZKSCAN3 and β-catenin expression in adenomas. ZKSCAN3 expression was correlated with that of β-catenin in colon adenomas (Figure 7D). Therefore, β-catenin-dependent ZKSCAN3 upregulation was observed in the early stages of carcinogenesis.

## 3. Discussion

The high expression of β-catenin caused by mutations in components of the WNT signaling pathway, such as the APC tumor suppressor, is one of the most frequent and early genetic changes that occur in colon cancer [35,36]. Cytoplasmic β-catenin translocates to the nucleus, where β-catenin/TCF complexes drive the expression of target genes, such as *c-Myc* and *cyclin D1* [37,38]. This promotes sustained cell growth and proliferation. Therefore, identification of the early transcriptional targets of the β-catenin/TCF complex is of paramount importance for understanding the molecular steps involved in colon carcinogenesis. We identified ZKSCAN3 as a downstream target of the WNT/β-catenin/TCF signaling pathway in colon cancer.

GO and KEGG pathway analyses indicated that DEGs in ZKSCAN3 KO cells were associated with cell cycle regulation, including the spindle assembly checkpoint, G1/S and G2/M transition, anaphase-promoting complex, and WNT signaling pathway. ZKSCAN3 could be a candidate for the Wnt signaling pathways involved in mitosis. qPCR and promoter assays using PNU and MSAB confirmed that ZKSCAN3 was transcriptionally regulated through the WNT/β-catenin/TCF signaling pathways in colon cancer cells. ZKSCAN3 is continuously upregulated in most human cancers; however, little is known about the mechanism underlying this upregulation [19,24,26,39]. The copy number of *ZKSCAN3* is increased in a subset of malignancies, including colon cancer, cervical cancer, bladder cancer, and prostatic cancer [21,24,25,26]. The ZKSCAN3 upregulation in various cancers could be attributed to this gene amplification. ZKSCAN3 overexpression in advanced cancers can be explained by an increase in gene copy variation; however, this model does not apply to early cancer precursor lesions or some advanced cancers without increased copy number. The regulation of ZKSCAN3 expression through the Wnt/β-catenin signaling pathway could partially explain the reason for ZKSCAN3 upregulation in adenomas without increased gene copies and in adenocarcinomas.

Dysplastic adenomas without cancerous transformation also have WNT/β-catenin-dependent ZKSCAN3 overexpression. Therefore, ZKSCAN3 could play a role, such as in disturbing chromosomal stability, in carcinogenic processes at the early stage of the adenoma-carcinoma sequence, considering that dysregulated WNT/β-catenin is the main and early driver of colon cancers [40]. Notable ZKSCAN3 overexpression is observed in the late stages of cancer or advanced cases with lymph node metastasis, indicating that ZKSCAN3 acts as a progression factor for malignant tumors [22]. ZKSCAN3 enhances the expression of a variety of target genes involved in cancer cell proliferation (cyclin D2), growth (IGF-2, integrin β4), metastasis (MMP26), and angiogenesis (VEGF) [23,27]. In contrast, aberrantly high expression of ZKSCAN3 was observed in dysplastic adenomas without ZKSCAN3 gene amplification, suggesting that ZKSCAN3 could play a role in both early carcinogenic processes and cancer progression. The additional data in this study clearly showed that ZKSCAN3 contributed to early tumorigenic events by inducing CIN.

In this study, we used GO, KEGG, and qPCR analyses to show that ZKSCAN3 induces CIN by inhibiting a set of spindle assembly checkpoint proteins involved in normal chromosomal segregation. CIN is one of the obvious and characteristic phenotypes that is closely related to cancer development and progression [41]. In terms of pathogenic mechanisms, CIN is closely linked to dysregulation or loss of components of the Wnt pathway [13]. APC is directly associated with mitotic spindle microtubules and is required for the spindle checkpoint to detect transiently misaligned chromosomes [42]. Therefore, the loss of functional *APC* by mutation leads to CIN [43]. In addition, the aberrant expression of β-catenin leads to CIN. The activation of the WNT pathway or β-catenin/TCF-mediated transcription increases CIN by dysregulation of G2/M progression or upregulation of conductin, respectively [12,14].

Here, we present a novel ZKSCAN3/MAD2L2 axis as a transcriptional target of the oncogenic Wnt/β-catenin pathways. ZKSCAN3, upregulated by β-catenin, was bound to the MAD2L2 promoter to inhibit MAD2L2 expression. This facilitated mitotic progression without the prolonged PT time. In contrast, ZKSCAN3 deficiency increased MAD2L2 expression, resulting in a delayed PT time. MAD2L2 rescue experiments support the hypothesis that the mitotic delay caused by ZKSCAN3 deficiency is mediated by MAD2L2 upregulation. Furthermore, FISH experiments showed that ZKSCAN3 overexpressing cells eventually exhibited chromosome loss or gain, as well as nuclear abnormalities such as micronuclei and nuclear blebs (Figure 6). This indicates that ZKSCAN3 induced both numerical and structural CIN. These findings suggest mechanisms underlying the induction of CIN by the aberrant activation of Wnt/β-catenin; it also explains how ZKSCAN3 participates in early carcinogenesis by hindering processes that ensure genetic stability, such as DNA repair [44].

MAD2L2 is a subunit of the shieldin complex, which regulates DNA repair at the damaged site and plays a critical role in the spindle assembly checkpoint (SAC) [45]. This prevents premature APC/C (CDC20) activation prior to anaphase onset, enabling faithful chromosome segregation [46]. MAD2L2 inhibits the growth of colorectal cancer by degrading nuclear receptor coactivator 3; it is a poor prognostic factor for colon cancer [47,48]. Defects in SAC can induce CIN, which confers genetic heterogeneity to initiate or accelerate the tumorigenic process [6,49]. However, somatic loss-of-function mutations in SAC, including that in MAD2L2, are rare and found in only a small fraction of cancers [48,50]. These findings imply that alternative mechanisms, such as suppression of expression rather than gene deletion, could be the major drivers of CIN in colon cancer. A recent study of NPC in relation to MAD2L2 expression supports our conclusions [51]. Coactivator-associated arginine methyltransferase 1 (CARM1) promotes MAD2 silencing by methylating the BAF155 subunit of the SW1/SNF complex on the MAD2L2 promoter; however, the regulatory mechanisms of MAD2L2 are yet to be elucidated [52]. Using qPCR array and ChIP analysis, we showed that ZKSCAN3 inhibited the MAD2L2 promoter activity, thereby inhibiting its expression; we identified ZKSCAN3 as a key transcriptional regulator of MAD2L2.

In conclusion, β-catenin-dependent ZKSCAN3 enhanced mitotic aberrations that contribute to CIN, mediated by a collective decrease in spindle assembly checkpoint proteins, including that of MAD2L2. Early upregulation of ZKSCAN3 in adenomas is not sufficient to drive the conversion of dysplastic adenoma to adenocarcinoma; however, this study indicates that ZKSCAN3 induces CIN and could play a crucial role in early stage carcinogenesis. Therefore, ZKSCAN3 may be used as an early tumor marker for colorectal cancer.

## 4. Materials and Methods

### 4.1. Cell Culture and Reagents

HCT116 colon cancer cells and a NCM365D normal colonic epithelial-derived cell line were purchased from the Korean Cell Line Bank (Seoul, Korea) and INCELL Corporation (San Antonio, TX, USA), respectively, and cultured in Dulbecco’s modified Eagle’s medium (DMEM) supplemented with 10% fetal bovine serum (FBS). The anti-ZKSCAN3 (ab223477), anti-MAD2L2 (ab180579), anti-GFP (ab290), anti-HA (ab9110), anti-α-tubulin (ab7291) and anti-V5 (ab27671) antibodies were purchased from Abcam (Cambridge, UK), and anti-β-catenin (sc7963), anti-GAPDH (sc47724) antibodies, and IgG (sc51993) were purchased from Santa Cruz Biotechnology, Inc. (Dallas, TX, USA). shRNAs for MAD2L2 and WNT proteins (WNT-1;120-17, WNT-3A; 315-20, and WNT-5A;645-WN-010) were obtained from PeproTech (Rocky Hill, NJ, USA) and R&D systems (Minneapolis, MN, USA). MSAB (SML1726) was purchased from Merck^®^ (Darmstadt, Germany). PNU-74654 (P0052), Cytochalasin B (C6762), CHIR 99,021 (SML1046), and all other reagents were purchased from Sigma-Aldrich, Inc. (St. Louis, MO, USA) unless otherwise specified. Cell proliferation was determined using an IncuCyte^®^ Live-Cell Imaging System (Essen Bioscience, Ann Arbor, MI, USA), and confluence was measured using IncuCyte 2016 B software.

### 4.2. Constructions of ZKSCAN3 Knockout Cell Lines

To develop ZKSCAN3 bi-allelic and mono-allelic mutant HCT116 cell lines (*ZK3*-KO and *ZK3*-KD cells, respectively), plasmids containing sgRNA sequences targeting *ZKSCAN3* exon1 and *Cas9* (pZKSCAN3-CRISPR/Cas9) were generated using pX458. sgRNA sequences are shown in Appendix A. pZKSCAN3-CRISPR/Cas9 were transfected into HCT116 cells, and the cells were cultured with puromycin (0.5 μg/mL) for 3 weeks to select stably transfected clones. After single-cell selection using puromycin, the genomic DNA (gDNA) sequences from each clone were analyzed. *ZK3*-KO and *ZK3*-KD cells were confirmed based on sequencing results (Appendix A).

### 4.3. RNA Sequencing and Informatics Analysis

Total RNA was isolated from biologically triplicated parent HCT116 cells (*ZK3*-WT) and *ZK3*-KO cells. Libraries were prepared for 150 bp paired-end fragments using the TruSeq Stranded mRNA LT Sample Prep Kit (Illumina, San Diego, CA, USA, #20020594). Total RNA (1 µg) was used for constructing single-stranded paired-end RNAs; sequencing was performed using NovaSeq 6000 system (Illumina). FASTQ files generated in this study or previously deposited as GSE95670 were uploaded to the Partek Flow server (Partek Inc.), and raw reads were quantified to hg19 (Ensembl Transcripts release 75) using Bowtie 2 aligner. Normalized read counts using transcripts per million (TPM) methods were statistically modeled using the ANOVA approach. The list of differentially expressed genes was analyzed using the gene ontology algorithm of DAVID (https://david.ncifcrf.gov, (accessed on 5 February 2021)) and the network analysis algorithm of STRING (https://string-db.org/cgi/input?sessionId=b9CWt347jcHu&input_page_show_search=off, (accessed on 5 February 2021)). RNA sequencing data generated in this study were deposited in the Gene Expression Omnibus (GEO) database, GSE172201.

### 4.4. Plasmids, Viruses, and Transfection

Plasmids for GFP-tagged ZKSCAN3 (pZK3-GFP) and adenovirus expressing HA-tagged ZKSCAN3 (Ad-ZK3/HA) were purchased from GeneCopoeia, Inc. (Rockville, MD, USA). Plasmids expressing GFP-tagged β-catenin (pCTNNB1), GFP-tagged TCF4 (pGFP-TCF4), pCI-H2B-RFP (pRFP-H2B), and GFP-tagged MAD2L2 (pDEST-MAD2L2-GFP) were obtained from Addgene (Cambridge, MA, USA). V5 tagged ZKSCAN3 in pcDNA (pcDNAV5-ZK3) was constructed as described previously [53]. Transfection was performed using Lipofectamine 2000 (Invitrogen, Carlsbad, CA, USA, #11668019) according to the manufacturer’s protocol.

### 4.5. RNA Isolation and Real-Time qPCR Analysis

Total RNA was isolated using the QIAzol reagent (Qiagen, Hilden, Germany, #79306), according to the manufacturer’s recommendation. Total RNA (2 µg) was reverse transcribed to cDNA using a reverse transcription kit (Thermo Fisher Scientific, Waltham, MA, USA, #4368814). qPCR was performed to determine the expression of genes, using SYBR Green PCR Master mix (Sigma-Aldrich, St. Louis, MO, USA, #S5193) and the StepOnePlus™ Real-Time PCR detection system (Step One Plus 2.02 software) with universal thermal cycling parameters according to the manufacturer’s protocol. Primer sequences used for qPCR are listed in Appendix A. GAPDH was used as an internal control to determine the reaction efficiency. The 2^−ΔΔCt^ method was used to determine the relative expression of the target genes [54].

### 4.6. Luciferase Assay

The ZKSCAN3 or MAD2L2 promoter sequences were amplified from human kidney genomic DNA using PCR as described previously [55]: the primers used are ZKSCAN3 promoter: 5′-TTCTGTATGATTGTTGGC-3′ (forward) and 5′-CCTAATTAGAATATGAATT-3′ (reverse); MAD2L2 promoter: 5′-AGCCCAGTGCCCAGCACTCG-3′ (forward) and 5′-GAAGCGGGGTTGGAATAAGAC-3′ (reverse). Promoter fragments were cloned in-frame into the pGL2 vector to generate pGL-ZK3 or pGL-MAD2L2. Plasmids were transfected into HCT116 cells for 24 h, followed by treatment of the cells with the indicated reagents; the cell lysates were obtained according to the manufacturer’s recommendations (Promega, Madison, WI, USA). Luciferase activity was measured in samples containing equivalent amounts of protein using a luminometer (PE; Applied Biosystems, Waltham, MA, USA) and luciferase assay reagents.

### 4.7. Live Cell Imaging

Cells were stably transfected with pRFP-H2B and cultured in a live-cell instrument (LCI, Gyeonggi-do, Republic of Korea). Fluorescent images were recorded using an inverted confocal microscope (ZEISS, Oberkochen, Germany), and transmitted light images were recorded using DIC optics.

### 4.8. Chromatin Immunoprecipitation (ChIP) Assay and EpiTect^®^ ChIP qPCR Array–Human Cell Cycle

The ChIP assay for TCF4 or ZKSCAN3 was performed using the ChIP Assay Kit (17–295, Sigma-Aldrich), as recommended by the manufacturer. Briefly, Ad-ZK3/HA or pGFP-TCF4 was transduced into HCT116 for 24 h and the cells were cross-linked by incubating with formaldehyde (final concentration 1%) for 10 min at 37 °C; the reaction was stopped by incubating with glycine (final concentration 125 mM) for 5 min. After washing, the collected cells were lysed in SDS lysis buffer (1% SDS, 50 mM Tris-HCl pH 8.1, and 10 mM EDTA) and the DNA was sheared into 200–1000 bp fragments using sonication. After centrifugation, supernatants containing the DNA–protein complexes were subjected to immunoprecipitation with anti-HA, anti-GFP, or anti-IgG antibodies overnight at 4 °C and with protein A for 1 h at 4 °C. The precipitates were washed five times and the DNA–protein complexes were eluted; crosslinks were reversed by adding 5 M NaCl (final concentration 200 mM) and heating at 65 °C for 4 h. DNA was recovered through phenol/chloroform extraction and ethanol precipitation. The eluted DNA fragments were analyzed through qPCR using the following primers: ZKSCAN3 promoter: 5′-TATGATTGTTGGCTTTTTTC-3′ (forward) and 5′-ATAAGGAAAACAATAAATAAA-3′ (reverse); MAD2L2 promoter: 5′-ACGTTGTGTTTTCTTCGCGC-3′ (forward) and 5′-AAGCGGGGTTGGAATAAGA-3′ (reverse); GAPDH promoter: 5′-GGACTCATGACCACAGTCCAT-3′ (forward) and 5′-GTTCAGCTCAGGGATGACCTT-3′ (reverse). The eluent containing DNA fragments pulled down with anti-HA (ZKSCAN3) was subjected to EpiTect^®^ ChIP qPCR array-Human Cell Cycle (QIAGEN, #334211) assay. Real-time qPCR was performed using ChIP PCR array plates, which included the promoters of 84 genes that are key to cell cycle regulation.

### 4.9. Cell Cycle Analysis, CBMN Assays, and Mitotic Index

The trypsinized cells were fixed with 70% cold ethanol for 30 min at 4 °C and washed twice with cold PBS. The cells were stained with propidium iodide solution (50 μg/mL) in PBS containing RNAse A (final concentration 0.5 μg/mL) at room temperature for 1 h. The samples were analyzed using a Cytomics FC500 flow cytometer (Beckman Coulter, Brea, CA, USA), and the results were analyzed using the CXP software (Beckman Coulter). CBMN assays with cytokinesis-block induced by cytochalasin B (CytB) were carried out, as described previously [56]. Briefly, HCT116 cells were cultured on coverslips for 24 h. The cells were treated with CytB (5 μg/mL) for 24 h, fixed with 4% paraformaldehyde for 15 min, and then stained with DAPI. The mitotic index was analyzed by counting mitotic cells with α-tubulin immune-staining.

### 4.10. Western Blotting and Immunofluorescence

Cell lysates were prepared from harvested cells using a commercially available kit (AKR-190, Cell Biolabs, San Diego, CA, USA), according to the manufacturer’s recommendation. Western blotting was performed as previously described [53]. Target genes or GAPDH expression using Western blotting was quantified using ImageJ software. The target gene/GAPDH ratio was compared to that of the control. For immunostaining, pretreated cells were fixed with 4% paraformaldehyde and incubated sequentially with primary antibodies at 4 °C overnight and secondary antibodies at 4 °C for 2 h. The cells were viewed under a Nomarski DIC-equipped inverted confocal microscope after nuclear staining with DAPI (Invitrogen, D1306).

### 4.11. Tissue Samples, Immunohistochemistry (IHC), and Evaluation

In total, 32 cases of hyperplastic polyps, 35 cases of adenomas, and 39 cases of adenocarcinomas arising from adenomas in FFPE tissue blocks were obtained from endoscopic biopsy or surgical specimens from the Department of Pathology, Kyung Hee University Hospital (Seoul, Korea) with the approval from Institutional Review Board (IRB #: 2019-12-060-001). Immunohistochemical staining for ZKSCAN3 was performed as described previously [26]. Staining was evaluated, as described previously [57]. Briefly, the staining intensity was graded as follows: 0, negative; 1, weak expression; 2, moderate expression; 3, strong expression and 4, a very strong expression. Immunohistochemical (IHC) scores were obtained by multiplying the percentage of positive cells by the staining intensity.

### 4.12. Quantitative PCR for Copy Number Aberration Detection

gDNA was extracted from FFPE tissues, and the gene copy number was detected, as described previously [58]. For each PCR assay, 5 ng of gDNA was used in the reaction mixture containing 2X TaqMan^TM^ genotyping master mix (Thermo Fisher Scientific), 20X TaqMan^TM^ Copy Number Assay, and nuclease-free water using StepOne^TM^ Real-Time PCR System (Applied Biosystems) as suggested by the manufacturer’s protocol (Thermo Fisher Scientific). PCR assays were performed in triplicate. A diploid gDNA was used as the control.

### 4.13. Fluorescence In Situ Hybridization

The fluorescence in situ hybridization (FISH) was performed as previously described [59]. For chromosomes 7, 12, and 17, we used the directly labeled green- or red-dUTP FISH probes (Empire Genomics, Buffalo, NY, USA), according to the manufacturer’s protocol. DAPI was used as the counterstain. The slides were examined using a confocal microscope with a 60X objective. Scoring and evaluation of FISH slides were performed manually by counting the target gene signals in at least 200 tumor cell nuclei per case.

## 5. Statistics

All experiments were performed at least thrice. Data are expressed as mean ± standard deviation of the mean (SD). Statistical analysis between comparable groups was performed using SPSS software, 13.0 (SPSS, Chicago, IL, USA). Normality distribution tests were performed using Shapiro–Wilk test. For parametric analysis, one-way ANOVA tests followed by Duncan’s post hoc tests were performed (Figure 6A and Figure 7A). For non-parametric analyses, Kruskal–Wallis tests followed by Dunn’s post hoc test were performed. For all statistical comparisons, statistical significance was set at *p* < 0.05. All statistical analyzes were performed by a statistician (B.G. Choe).

## Figures and Tables

**Figure 1 ijms-23-09302-f001:**
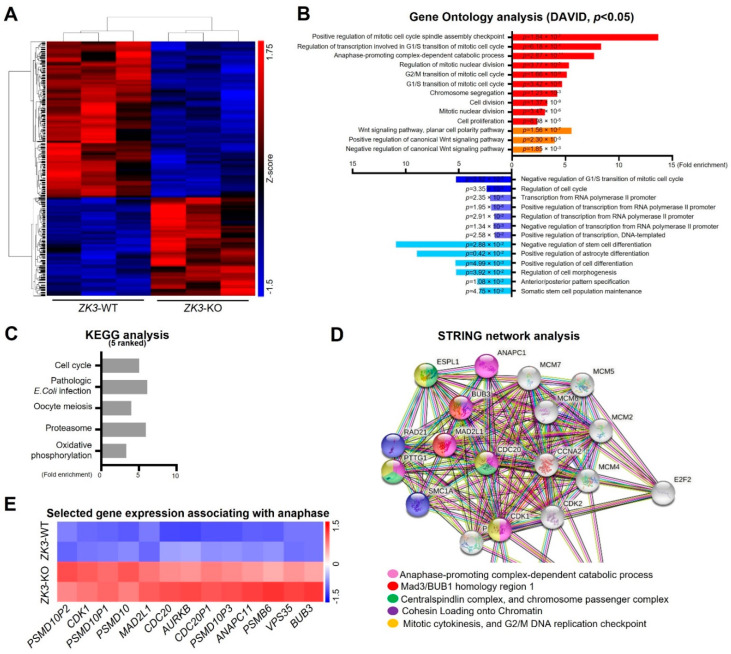
Transcriptome analysis of ZK3-KO cells versus reference cells. (**A**) Hierarchical clustering heat-map for DEGs of the ZK3-KO HCT116 and parental HCT116 cells. (**B**) GO analysis for biological processes. Red bars represent categories generated from upregulated genes, and blue bars are those from the downregulated genes. *p*-values are indicated. (**C**) Top 5 KEGG pathways in pathway enrichment analysis of the DEGs. (**D**) STRING network analysis of cell cycle-associated genes classified by GO analysis. (**E**) Heat-map plot comparing the transcription levels of genes involved in the regulation of anaphase, based on the RNA sequencing results. GO, gene ontology; DEGs, differentially expressed genes; KEGG, Kyoto Encyclopedia of Genes and Genomes.

**Figure 2 ijms-23-09302-f002:**
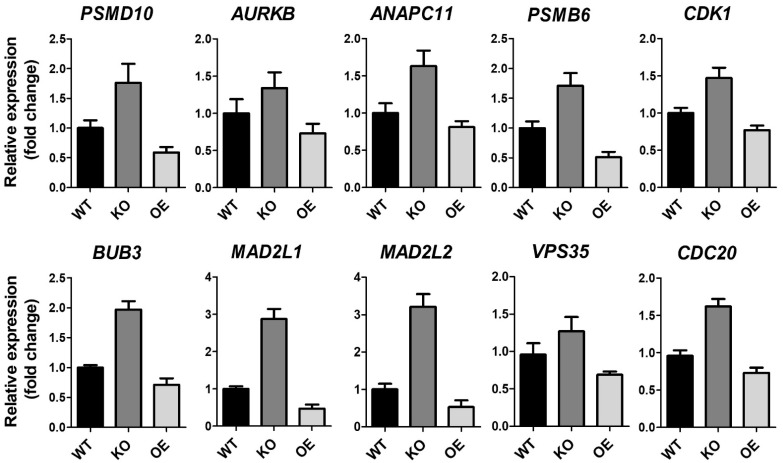
RNA expression of ZK3-KO and reference cells. Validation of anaphase-associating genes from HCT116 (WT), ZK3-KO (KO), and ZKSCAN3-overexpressing (OE) HCT116 cells transduced by Ad-ZK3/HA, using qRT-PCR. Data from three independent experiments are shown as the means ± SD. Statistical significance was determined using the Kruskal–Wallis test (*p* < 0.01).

**Figure 3 ijms-23-09302-f003:**
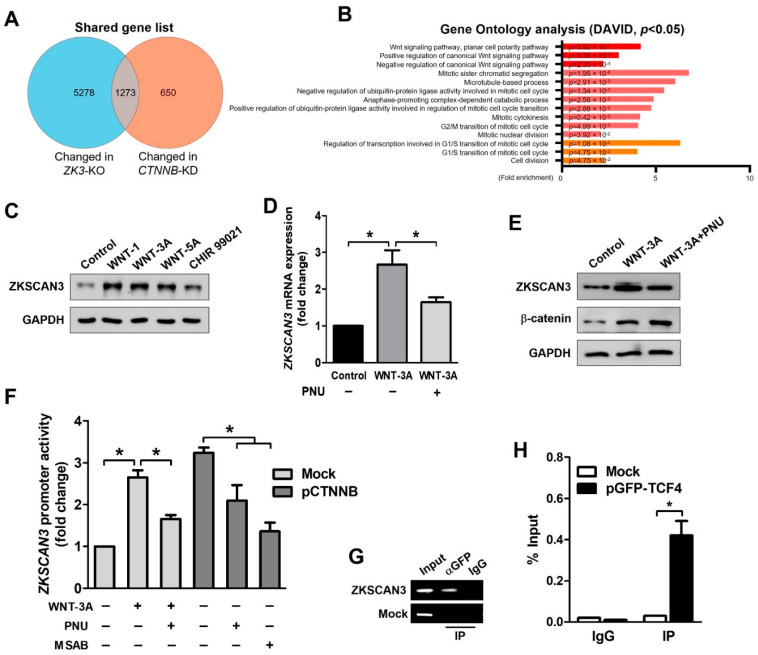
WNT signals upregulate ZKSCAN3 transcription through a β-catenin-dependent pathway. (**A**) Venn diagram comparing the DEGs of ZK3-KO cells with previously reported results using the CTNNB1-KD HCT116 cell line. (**B**) GO analysis using selected genes, which were altered in both the RNA sequencing results. *p*-values are indicated. (**C**) HCT116 cells were treated with WNT-1 (50 ng/mL), WNT-3A (100 ng/mL), WNT-5A (50 ng/mL), or CHIR 99,021 (3 μM) for 24 h, and cell lysates were subjected to Western blotting using the indicated antibodies. (D, E) HCT116 cells were treated with WNT-3A (100 ng/mL) for 24 h in the presence or absence of 40 μM PNU and cells were harvested. Extracted mRNA and cell lysates were subjected to qPCR against ZKSCAN3 (**D**) or Western blotting using the indicated antibodies (**E**), respectively. The qPCR data obtained in triplicate are indicated as fold changes of the ZKSCAN3 mRNA levels relative to that in the untreated control cells. Statistical significance was determined using the Kruskal–Wallis test (* *p* < 0.05). (**F**) Cells were transfected with both pGL-ZK3 and pCTNNB1 or empty vector for 24 h, followed by WNT-3A (100 ng/mL) treatment with or without PNU (40 μM) or MSAB (10 μM) for 24 h. The luciferase activity was determined, as described above. Data are expressed as fold change over that in the control cells; they were obtained from three independent experiments and are shown as means ± SD. Statistical significance was determined using the Kruskal–Wallis test (* *p* < 0.05). (**G**) ChIP of transfected cells. Cells were transfected with the pGFP-TCF4 vector. ChIP was performed using an anti-GFP antibody or IgG, followed by PCR amplification of the ZKSCAN3 promoter. (**H**) ChIP-qPCR assay in transfected cells. Data are represented as means ± SD of triplicate measurements. Statistical significance was determined using the Wilcoxon test (* *p* < 0.05).

**Figure 4 ijms-23-09302-f004:**
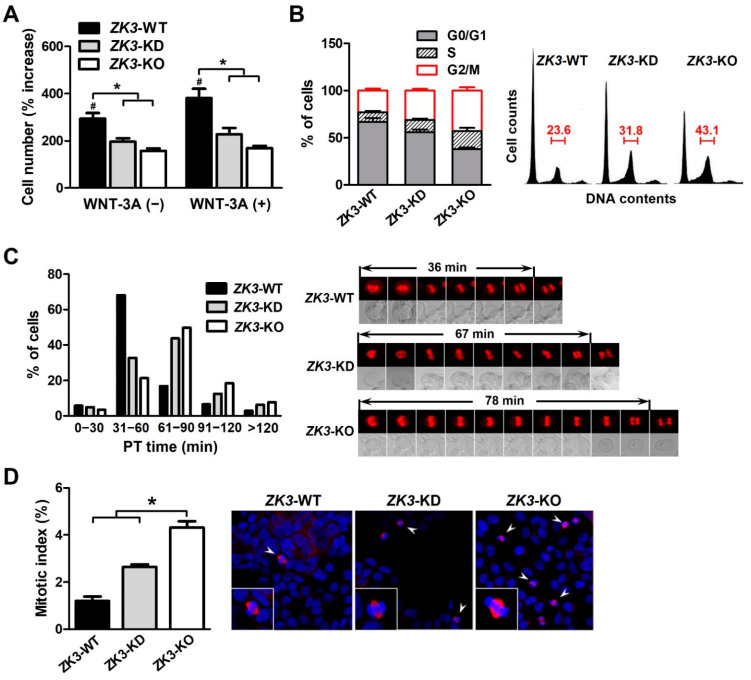
ZKSCAN3 deficiency delays the progression of mitosis. (**A**) Cell proliferation rate was determined in the presence or absence of WNT-3A through sequential monitoring with Incucyte. The cell count data were acquired 6 days after initial seeding. Data are expressed as mean ± SDs (*, # *p* < 0.05). (**B**) Cell cycle profiles were determined using flow cytometry after staining with propidium iodide. Histogram data were obtained from three independent experiments and are expressed as mean ± SDs. Representative flow cytometric results are shown in the right panel. (**C**) Time-lapse analysis of ZK3-WT, ZK3-KD, and ZK3-KO cells. Cells stably expressing histone 2B-RFP were cultured on a live cell chamber instrument and continuous fluorescent images were taken. The graph presents the percentage of cells at different time points in the time taken from prometaphase to telophase (PT time). A minimum of 50 cells per sample was observed. Representative images with mean PT times are shown in the right panel. (**D**) Mitotic indices were determined by counting cells, which had pairs of duplicated chromatids aligned to the center of the dividing cell (arrowhead). Unsynchronized ZK3-WT, ZK3-KD, and ZK3-KO cells were immunostained with anti-α-tubulin antibody, and nuclei were stained with DAPI. Individual cells are magnified in the box. A minimum of 200 cells were counted on three slides. The histogram displays the mean ± SD of triplicate measurements (left panel). Representative images are shown in the right panel. Statistical significance was determined using the Kruskal–Wallis test (* *p* < 0.01).

**Figure 5 ijms-23-09302-f005:**
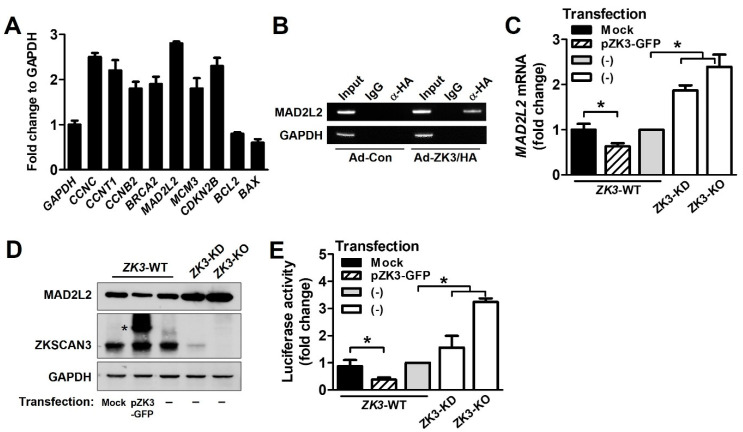
ZKSCAN3 acts as a transcriptional repressor of MAD2L2. (**A**) Chip assay and qPCR array were performed using EpiTect^®^ ChIP qPCR array Human Cell Cycle (QIAGEN). HCT116 cells were transduced with Ad-ZK3/HA or Ad-Con for 24 h and the crosslinked lysates were sonicated. The sonicated lysates were immunoprecipitated with anti-HA antibody or IgG and then with protein A. The eluted precipitates were applied to a qPCR array (QIAGEN). Representative qPCR results from the array are presented. Data are expressed as mean ± SD of triplicate experiments; GAPDH expression was used as the internal control. (**B**) Cells were infected with Ad-Con or Ad-ZK3/HA for 24 h, treated with formaldehyde, and sonicated. The cross-linked DNA–protein complex was immunoprecipitated using anti-HA (α-HA) antibody or IgG followed by PCR amplification for MAD2L2 promoter. Immunoprecipitates obtained using IgG were used as the negative controls. Input chromatin (Input) refers to sonicated chromatin before immunoprecipitation. (**C**) Real-time qPCR was performed to evaluate the expression of MAD2L2 using mRNA extracted from the indicated cells. Data obtained from three independent experiments were used to represent a fold increase in mRNA levels. Statistical significance was determined using the Kruskal–Wallis test (* *p* < 0.05). (**D**) Western blotting was performed to evaluate the expression of MAD2L2 and ZKSCAN3 proteins; GAPDH was used as the loading control. *, ectopic ZKSCAN3 (**E**) The indicated cells were transfected with pGL-MAD2L2. The luciferase activity was determined. Data are expressed as fold change in expression compared to that in control cells, from three independent experiments; they are expressed mean ± SD. Statistical significance was determined using the Kruskal–Wallis test. (* *p* < 0.01).

**Figure 6 ijms-23-09302-f006:**
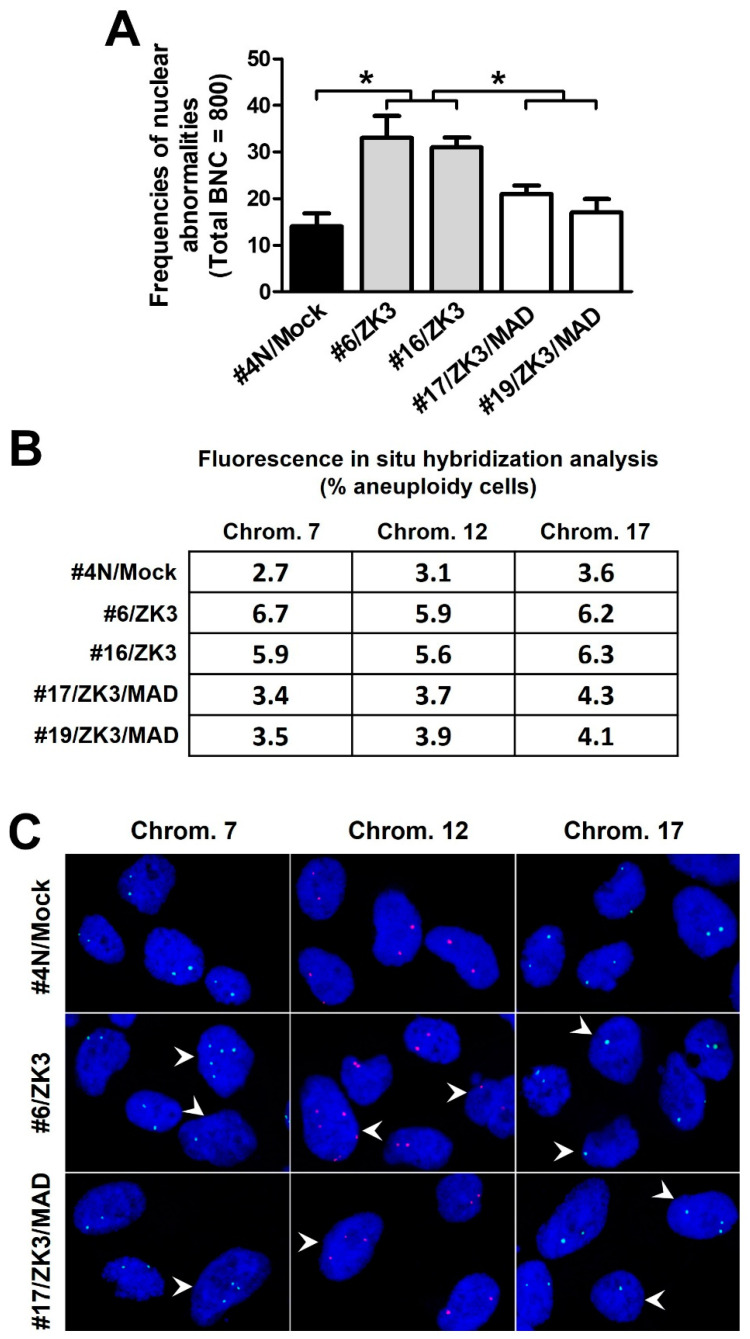
ZKSCAN3 induces chromosomal instability by suppressing MAD2L2 expression. (**A**) CBMN assays for the indicated cells derived from single cells after 40−50 generations. A total of 800 binucleated cells (BNC) were scored for nuclear abnormalities. Data are expressed as mean ± SD. * Statistical significance was determined using one-way ANOVA followed by Duncan’s test (* *p* < 0.01). (**B**) FISH analysis of interphase cells using CEP probes. Average 300−500 cells were scored for each cell line; data are expressed as the percentage of aneuploid cells. (**C**) Fluorescence microscopy images of interphase FISH for indicated cells using probes for chromosomes 7, 12, and 17. Aneuploid cells (arrowhead). Optical magnification, ×600. CBMN, cytokinesis-block micronucleus.

**Figure 7 ijms-23-09302-f007:**
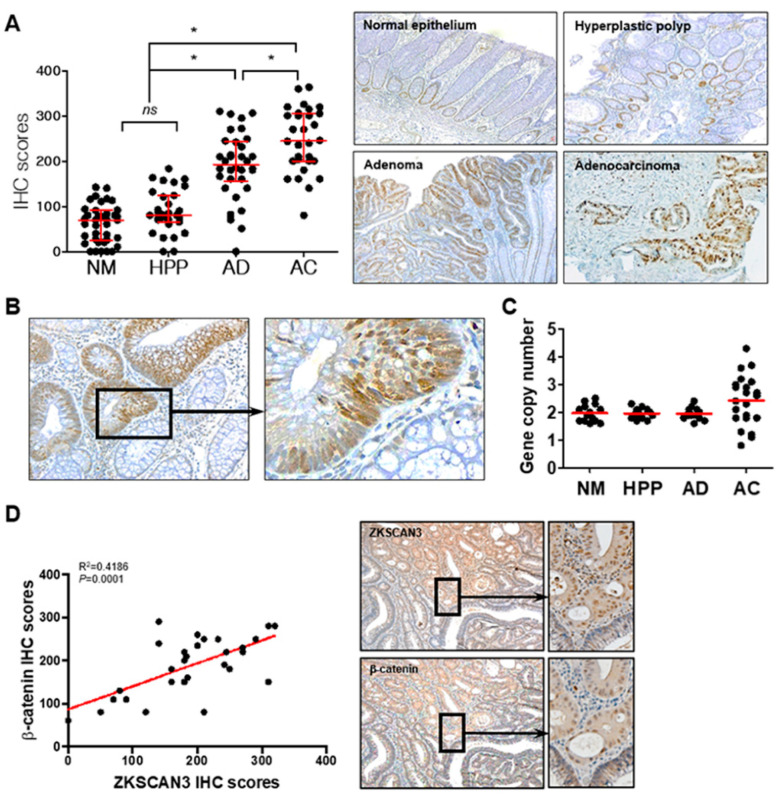
ZKSCAN3 upregulation in the early stage of the adenoma–carcinoma sequence. (**A**) Distribution of IHC scores in the normal colonic epithelium (NM; *n* = 38), non-neoplastic hyperplastic polyps (HPP; *n* = 31), adenomas (AD; *n* = 34), adenocarcinomas (AC; *n* = 30), and their representative microscopic images (right panel). Optical magnification, ×100. Red bars represent the median with quartiles of the IHC scores. Statistical significance was determined using one-way ANOVA followed by Duncan’s test (* *p* < 0.01). (**B**) Representative ZKSCAN3 IHC images of adenoma with dysplasia. Optical magnification, ×200 (left panel) and ×400 (right panel). (**C**) Gene copy numbers of normal epithelium (*n* = 14), non-neoplastic hyperplastic polyps (*n* = 16), dysplastic adenomas (*n* = 14), and adenocarcinomas (*n* = 22). (**D**) Correlation between IHC scores of ZKSCAN3 and β-catenin. Data were obtained from 29 cases of adenomas and analyzed through Pearson correlation (*p* < 0.01). Representative IHC images are shown in the right panel. Optical magnification, 100 (middle panel) and ×200 (right panel). ns, not significant.

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
