# Peer review of "Role of the WNT/β-catenin/ZKSCAN3 Pathway in Regulating Chromosomal Instability in Colon Cancer Cell lines and Tissues"

_ijms, 2022, doi:10.3390/ijms23169302_

Round 1

Reviewer 1 Report

Paper titled (ZKSCAN3 Regulated by the Canonical WNT/β-catenin Pathway Induces Chromosomal Instability in Colon Cancer Cells) by Cho et al. discussed the role of  the Canonical WNT/β-catenin Pathway in regulating ZKSCAN3 and Chromosomal Instability in Colon Cancer Cell lines & biopsies.

1- title does not indicate anything about biopsies

2- Aim of the work did not mention that pathway will be studied in colon cancer (NOT other cancers) & also conclusion.

3- Figure 1 F is not clear & need to be divided into panels (may be in a separate figure)

4- Figure 6: IHC should not be quantified by scoring. Instead image analysis for area of immunostaining can be correct

5- Avoid very long paragraphs

6- how PCR was quantified? what refernce for this?

7- Methods: How WB was quantified ? what house keeping protein?

Reviewer 2 Report

Authors have answered all my concerns.

Author Response

We appreciate your kind comments and efforts.

Round 2

Reviewer 1 Report

Thanks

This manuscript is a resubmission of an earlier submission. The following is a list of the peer review reports and author responses from that submission.

Round 1

Reviewer 1 Report

it is an interesting manuscript. please include the potential clinical impact of your results

Author Response

Response to Reviewer 1 Comments

    We are thankful to the reviewer’s comments regarding our paper. We fully agree with the reviewer’s comments, and we have corrected and responded to it as follows.

Point 1: it is an interesting manuscript. please include the potential clinical impact of your results

Response 1: We added a sentence describing the clinical impact of our results in conclusion section (line 444).

Reviewer 2 Report

Paper titled (ZKSCAN3 Regulated by Canonical WNT/β-catenin Pathway 2 Induces Chromosomal Instability in Colon Cancer Cells) by Cho et al. 

1- Abstract is not fine and should contain some numerical values.

2- Stat analysis: Authors have to check the normality of distribution of the results by a suitable post hoc test (such as Shapiro-Wilk test or K-S test) before deciding to choose certain ANOVA. If the normality test indicated normal dist of the data, so use one-way ANOVA, if not, use non parametric ANOVA. In all cases choose a suitable post-hoc test

3- Authors should give the source of chemicals, kits and antibodies completely and consistently (code, company, town, state and country) & version for software

4- Resolution of figure 1 is poor & better spletted into 2 separate figures

5- In each illustration mention the type of the presented data & the statistical test applied for analysis

6- Methods in general lacks references at many occasions,

7- what is the refreence for PCR delta CT method? line 481

8- The method and software used for quantification in WB analysis should be added.

 9- Data should be presented as mean+-SD (not SE) this is as authors do not cover the universe for this study.

10- Mention n in figure legends

Reviewer 3 Report

The article of Park and colleagues “ZKSCAN3 regulated by canonical Wnt/B-catenin parthway induces chromosomal instability in colon cancer cells” is interesting, well described, and the majority of the experiments performed and controlled for well. Although, there are a few issues they should address before publication.

Major

·       - Wnt5a is primarily a non-canonical ligand – yet induces increased ZKSCAN3 expression. Authors should explain this in regards to their claim it is regulated through canonical mechanisms.

·        -Authors should address whether Wnt activation suppresses MAD2L2 expression?

·      - No statistical analysis presented on the data described in Figure 6.

·        - No actual pearson, or spearman correlation described (linear slope)

·        - Is the correlation between total beta-catenin staining, or just nuclear (which would be more appropriate)

Minor points.

Typographical errors.

·       -Many uses of CTNNB, should be CTNNB1

·        - Hct116 à HCT116

Round 2

Reviewer 2 Report

Unfourtntely, authors did not respond to my previous questions and did not check the normality of data distribution before they apply t test!! also scoring IHC is not appropriate and should be image anlayzed insrtaed - if we use scoring, we cannot present data as mean+-SD, should be median & quartiles.

Is it easy to change SE to SD and keep data as is ?

Reviewer 3 Report

Authors have addressed all my comments 

Author Response

We deeply appreciate for your review and advice.